# Effects of Astaxanthin on Growth Performance, Gut Structure, and Intestinal Microorganisms of *Penaeus vannamei* under Microcystin-LR Stress

**DOI:** 10.3390/ani14010058

**Published:** 2023-12-22

**Authors:** Guolin Song, Yingcan Zhao, Junhao Lu, Zhe Liu, Jinqiang Quan, Lirui Zhu

**Affiliations:** College of Animal Science and Technology, Gansu Agricultural University, Lanzhou 730070, China; 18419613067@163.com (Y.Z.); 18438615011@163.com (J.L.); liuz@gsau.edu.cn (Z.L.); quanjq@gsau.edu.cn (J.Q.); zhulr@gsau.edu.cn (L.Z.)

**Keywords:** *P. vannamei*, MC-LR, astaxanthin, growth performance, intestinal microbiota

## Abstract

**Simple Summary:**

The frequent occurrence of cyanobacterial blooms in shrimp ponds has emerged as a significant challenge for *Penaeus vannamei* aquaculture. This study aimed to investigate the protective effect of astaxanthin (AX) on *Penaeus vannamei*, which is harmed by Microcystin-LR (MC-LR), through analyzing the former’s growth performance, intestinal structure, and intestinal microbiota. The results demonstrated that AX had positive impacts on the growth performance and resistance of *Penaeus vannamei* against MC-LR through the regulation of intestinal microbiota composition. It can be considered as a suitable additive in *Penaeus vannamei* feed to mitigate the detrimental effects of MC-LR on shrimp.

**Abstract:**

Microcystin-LR (MC-LR) are biologically active cycloheptapeptide compounds that are released by cyanobacteria during water blooms and are extensively found in aquatic ecosystems. The *Penaeus vannamei* is a significant species in global aquaculture. However, the high level of eutrophication in aquaculture water frequently leads to outbreaks of cyanobacterial blooms, posing a significant threat to its sustainable cultivation. Astaxanthin (AX) is commonly utilized in aquaculture for its physiological benefits, including promoting growth and enhancing immune function in cultured organisms. This study aimed to examine the protective effect of astaxanthin on *P. vannamei* exposed to microcystin-induced stress. The experiment consisted of three groups: one group was fed formulated feed containing MC (100 μg/kg), another group was fed formulated feed containing MC (100 μg/kg) + AX (100 mg/kg), and the third group was fed basic feed (control group). After 15 days of feeding, the specific growth rate (SGR) was significantly higher in the MCAX group (2.21% day^−1^) compared to the MC group (0.77% day^−1^), and there was no significant difference between the MCAX group (2.21% day^−1^) and the control group (2.24% day^−1^). Similarly, the percent of weight gain (PWG) was also significantly higher in the MCAX group (14.61%) compared to the MC group (13.44%) and the control group (16.64%). Compared to the control group, the epithelial cells in the MC group suffered severe damage and detachment from the basement membrane. However, in the MCAX group, although there was still a gap between the intestinal epithelial cells and the basement membrane, the overall intestinal morphology was slightly less impaired than it was in the MC group. The analysis of the intestinal microbiota revealed a significant disparity in the community composition (chao 1 and ACE) between the MC and MCAX groups. When comparing the various bacterial genera, the MC group exhibited an increase in *Vibrio* abundance, whereas the MCAX group showed a decrease in both *Shewanella* and *Vibrio* abundance. The results indicate that AX has a positive impact on the growth performance and resistance of *P. vannamei* against MC by regulating the composition of the intestinal microbiota. AX can be utilized to mitigate the detrimental effects of MC in aquaculture practices. This function could be attributed to the role of AX in preserving the structural integrity of the intestinal mucosa and regulating the composition of the intestinal microbiota.

## 1. Introduction

The species *P. vannamei* is native to the eastern coast of the Pacific Ocean, specifically the central regions of Peru and Ecuador. Known for its adaptability, fast growth, and high meat yield, it is considered one of the primary shrimp species cultivated worldwide [1]. Recently, there has been rapid development in *P. vannamei* aquaculture, which has become a significant contributor to the aquaculture economy in many countries [2]. The aquaculture model is characterized by high density, intensification, and scale, leading to increased bioburden and elevated input requirements for the aquaculture’s water. The decomposition of a significant quantity of excreta and surplus feed in the body of water leads to the release of substantial amounts of nitrogen and phosphorus nutrients. This, in turn, contributes to the worsening eutrophication of the water body, resulting in significant harm to the health of shrimp farming using *P. vannamei*. Studies have demonstrated that a water body’s eutrophication can result in the production of a significant quantity of microcystins (MCs). These MCs are secondary metabolites generated by cyanobacteria, specifically *Microcystis* sp. and *Cichlidium* sp., during bloom outbreaks [3,4]. Furthermore, they are among the most prevalent cyanobacterial toxins found in freshwater. MCs are a group of cyclic heptapeptides [5]. Over 100 variants of MCs have been reported, including MC-LR [6]. MCs can accumulate in the tissues of aquatic organisms found in freshwater lakes, such as snails, shrimps, and fish [7,8,9]. MCs can impact the growth rate of aquatic animals, leading to a decrease in shrimp growth and causing pathological changes in shrimp organs, including the hepatopancreas, intestines, and gonads [10]. Additionally, MCs can alter the composition of intestinal microorganisms in shrimp, resulting in changes in composition [11,12]. Therefore, enhancing the ability of *P. vannamei* to counteract the toxicity of MCs prior to cyanobacterial outbreaks holds significant practical importance.

Astaxanthin is a carotenoid that is not derived from vitamin A and is known for its strong antioxidant properties [13,14]. Astaxanthin is widely found in nature [15]. Studies have confirmed that astaxanthin promotes the growth, survival, and coloration of cultured subjects [16]. Additionally, it possesses strong antioxidant, antitumor, and immune-enhancing physiological functions [17,18]. For example, the addition of astaxanthin significantly enhances the growth, survival, and stress resistance of black tiger prawn (*Penaeus monodon*) [19]. Adding astaxanthin to the feed enhances the growth performance, survival, and stress tolerance of kuruma shrimp (*Marsupenaeus japonicus*) [20].

In this study, MC-LR and AX were added to the feed. After breeding, we evaluated the impact of AX on the intestinal microbial population of *P. vannamei* and determined its protective effect on the intestine following an MC-LR challenge. The findings of this study will offer valuable insights into the protective effect of AX against MC-LR in shrimp and enhance our understanding of the protective mechanisms of AX in the intestine of *P. vannamei* affected by MC-LR.

## 2. Materials and Methods

### 2.1. Preparation of AX and MC-LR

The MC-LR was produced by Shanghai Yuanye Biotechnology Co., Ltd. (Shanghai, China). The purity of the MC-LR was 95%. The AX was manufactured by BASF Europe, and contains 11% AX. The feed (crude protein 42%, crude fat 4%, crude fiber 5%, ash 16%) for *P. vannamei* was selected from the commercially available *P. vannamei* compound feed produced by Jieyang Hai Da Group. In every kilogram of feed, 100 mg/kg of AX and 100 μg/kg of MC-LR were added. These concentrations were determined based on pre-tests. MC-LR and AX were dissolved separately in water to prepare a 3% stock solution. Then, according to the experimental design (Table 1), the MC-LR and AX solution was sprayed onto the basal feed. The control diet was sprayed with water without the addition of AX and MC-LR. The feed was naturally dried under normal conditions and stored at 4 °C.

### 2.2. Shrimp and Rearing Conditions

The test feeds were administered three times daily, at 8:00, 12:00, and 18:00, during both the transient and experimental periods. The daily feeding rate corresponded to 3% of the shrimp’s body mass. Measurements of salinity, water temperature and pH, as well as dissolved oxygen, ammonia, and nitrite nitrogen concentrations, were taken at 8:30, 12:30, and 18:30. Throughout the experiment, the dissolved oxygen level was maintained within the range of (9.0 ± 0.5) mg/L, salinity was 4‰ ammonia nitrogen concentration was within the range of (0.10 ± 0.05) mg/L, pH was within the range of 8.5 ± 0.2, and nitrite nitrogen concentration was within the range of (0.01 ± 0.001) mg/L.

### 2.3. Experimental Design

A total of 1800 *P. vannamei* with a consistent genetic background, robust body mass, and an average body mass of approximately 0.5 ± 0.01 g were selected. The shrimps were randomly divided into three groups: control group, MC group, and MC + AX group. Each group had three replicates of 200 shrimps. The shrimps were kept in nine nets measuring 90 cm × 60 cm × 90 cm, with a mesh diameter of 0.2 cm. After a 7-day period of acclimation, the shrimps entered the experimental phase. The control and treatment (MC) groups were provided with a basal diet, while the treatment group (MCAX) received a diet containing 100 mg/kg of AX. The MCAX group was fed this diet for a duration of 30 days. At the conclusion of the 30th day, the control group maintained their consumption of the basal diet. The MC group received a diet enriched with MC-LR (100 μg/kg), while the MCAX group received a diet containing both MC-LR (100 μg/kg) and AX (100 mg/kg) for a duration of 15 days.

### 2.4. Sample Collection and Growth Performance

At the conclusion of the feeding trial, ten shrimps were randomly selected from each tank, weighed, and then aseptically sacrificed in an ice bath. The percent of weight gain (PWG, %), specific growth rate (SGR), and survival rate (%) of the *P. vannamei* were determined using the following formula:

Percent weight gain (PWG, %) = (final weight − initial weight)/initial weight × 100.

Specific growth rate (SGR, % day^−1^) = ln (final weight/initial weight)/days of experiment.

Survival rate (%) = (final number of shrimps)/(initial number of shrimps) ×100.

Intestine tissue was rapidly frozen in liquid nitrogen and stored at −80 °C for Illumina sequencing and real-time PCR. Intestines were fixed in neutral 10% formalin and embedded in paraffin. Sections of 5 μm thickness were stained with hematoxylin and eosin and examined with a light microscope.

### 2.5. Histopathological Analysis of the Intestine

The intestines of three shrimps selected from each tank were dissected and fixed in the solution at the end of the experiment. The intestines were fixed in 10% formalin for 24 h, dehydrated in a sequential series of alcohol (50–95%) and embedded in paraffin. Tissue sections (5 μm thick) were stained with hematoxylin and eosin, after which the intestinal structure was analyzed by microscopy, using an Olympus CKX41 microscope (Tokyo, Japan).

### 2.6. DNA Extraction and Amplification

The intestines of six shrimps selected from each tank were dissected and mixed for one microbial sample, and each group included three replicate microbial samples. All of the samples were immediately snap-frozen in liquid nitrogen for further analysis. The total genomic DNA was extracted using the MagPure Soil DNA LQ Kit (Magan, Shanghai, China) according to the manufacturer’s instructions. The concentration and integrity of the DNA were measured using a NanoDrop 2000 (Thermo Fisher Scientific, Waltham, MA, USA) and agarose gel electrophoresis. The extracted DNA was stored at −20 °C until further processing. This extracted DNA served as the template for PCR amplification of the bacterial 16S rRNA genes using barcoded primers and Takara Ex Taq (Takara, Japan). For the analysis of bacterial diversity, the V3–V4 variable regions of the 16S rRNA genes were amplified using universal primers 343F (5′-TACGGRAGGCAGCAG-3′) and 798R (5′-AGGGTATCTAATCCT-3′) [21] for the V3–V4 regions.

### 2.7. Library Construction and Sequencing

The quality of the amplicon was assessed through agarose gel electrophoresis. The PCR products were purified using Agencourt AMPure XP beads (Beckman Coulter, Jersey City, NJ, USA) and subjected to an additional round of amplification. After being purified once again with the AMPure XP beads, the final amplicon was quantified using the Qubit dsDNA Assay Kit (Thermo Fisher Scientific, Waltham, MA, USA). The concentrations were subsequently adjusted for sequencing. The sequencing was conducted on an Illumina NovaSeq 6000 platform using 250 bp paired-end reads (Illumina Inc., San, CA, USA; OE Biotech Company; Shanghai, China).

### 2.8. Bioinformatic Analysis

The paired-end reads underwent preprocessing using Cutadapt software to identify and remove the adapter. After trimming, the paired-end reads were filtered for low-quality sequences, denoised, merged, and had chimera reads detected and removed using DADA2 [22] with the default parameters of QIIME2 (2020.11) [23]. Finally, the software generated the representative reads and the ASV abundance table. The representative read for each ASV was selected using the QIIME2 package. All representative reads were annotated and compared to the Silva database (Version 138) using a q2-feature-classifier with the default parameters.

QIIME2 software was used for alpha and beta diversity analysis. The microbial diversity in samples was estimated using the alpha diversity, which includes the Chao1 index [24], ACE, Shannon index [25] and Simpson index. The binary Jaccard distance matrix performed by the R package(v 3.2.0) was used for the binary Jaccard principal coordinates analysis (PCoA) to estimate beta diversity. Then, the R package(v 3.2.0) was used to analyze the significant differences between different groups using an ANOVA statistical test. The linear discriminant analysis effect size (LEfSe) method was used to compare the taxonomy abundance spectrum.

### 2.9. Statistical Analysis

The value of each variable was expressed as the mean ± SD. The degrees of significance of the differences among the three groups were determined with Student’s *t*-test if the values were distributed normally and the variances were homogeneous. Statistical significance was determined by one-way analysis of variance (ANOVA) followed by Duncan’s multiple range test (SPSS ver 22.0). All *p*-values < 0.05 were considered to be significant.

## 3. Results

### 3.1. Growth Performance

The growth performance levels of *P. vannamei* treated with three different diets are shown in Table 2. There were significant differences in SGR and PWG between the groups. SGR in the MC group was lower than in the control group and the MCAX group. The PWG of the control group was higher than in the group fed with MC and MCAX (*p* < 0.05). However, the survival rates of three treatment groups were not significantly different (*p* > 0.05).

### 3.2. Intestinal Histological Alterations

We used H&E staining to monitor the intestinal epithelial morphology of the *P. vannamei*. In the control group, the intestinal epithelial cells were closely arranged and closely connected to the basement membrane (Figure 1A). The epithelial cells in the MC group completely detached from the basement membrane and were severely destroyed compared to the control (Figure 1B). In the MCAX group, there was a gap between the intestinal epithelial cells and the basement membrane, and the epithelial cell layer was separated from the basement membrane to some extent, compared to control group, while the overall intestinal morphology was slightly less impaired than in the MC group (Figure 1C).

### 3.3. Intestinal Microbiota Changes

#### 3.3.1. Richness and Diversity

The data volume of the raw reads under sequencing is distributed between 78,003 and 81,845, the data volume of clean tags after quality control is distributed between 67,403 and 75,401, and the data volume of valid tags after removing chimeras from clean tags is distributed between 59,995 and 72,838. A statistically significant difference (*p* < 0.05) was observed in both the Chao1 (298.99 ± 87.09, versus 100.68 ± 36.93, versus 359.13 ± 113.18) and ACE (299.07 ± 87.49, versus 100.75 ± 37.04, versus 359.58 ± 113.61) indices. However, no significant difference (*p* > 0.05) was found in the Simpson (0.90 ± 0.03, versus 0.90 ± 0.03, versus 0.80 ± 0.11) and Shannon (5.59 ± 0.83, versus 4.45 ± 0.41, versus 4.75 ± 0.96) indices among the three groups (Figure 2A–D). Intergroup analysis of alpha diversity indicated that the administration of MC significantly affected the gut microbial richness of shrimps but had no impact on microbial diversity. The unweighted PCoA plots showed that most samples from the MCAX group clustered separately from the MC group, which was consistent with the results of UPGMA analysis. This suggests significant differences in the principal compositions of gut microbiota among the three groups (Figure 3A,B).

The Venn diagram was used for community analysis to ascertain which species were common and which were unique among the different treatment groups (Figure 4). The results indicated that there were significant differences in the composition of microbiota between the three groups (Figure 4). A total of 98 ASV were ASV common to the control group and the experimental group. The special bacterium of the MCAX group (418 ASV) was compared with that of the MC group (128 ASV) and the control group (399 ASV).

#### 3.3.2. Intestinal Microbial Composition

To further explore the composition of the bacterial community richness in each group, the abundance of the intestinal microbiota was calculated in each group at the phylum and genus levels. As shown in Figure 5A, the primary intestinal microbiota in all groups were Proteobacteria, Firmicutes, Actinobacteriota, Bacteroidota, and Cyanobacteria at the level of phylum, among which Proteobacteria was the dominant phylum. The relative abundance of Proteobacteria and Cyanobacteria in the MC group increased significantly compared with the control group, while the relative abundance of Firmicutes and Actinobacteriota decreased. The relative abundance of Proteobacteria, Bacteroidota, and Cyanobacteria in the MCAX group decreased compared with that of the the MC group, while the relative abundance of Firmicutes and Actinobacteriota increased.

At the genus level, the relative abundance of the intestinal microbiota in each group is shown in Figure 5B. The abundance of the intestinal microbiota in the experimental group changed greatly at the genus level compared with that of the control group. The relative abundance of Candidatus_Bacilloplasma, PeM15, and Candidatus_Berkiella in the MCAX group increased compared with levels in the control group, while the relative abundance of Shewanella and Vibrio decreased. The relative abundance of Vibrio and Gemmobacter in the MC group increased compared with levels in the control group, while the relative abundance of Rhodobacter, Candidatus_Bacilloplasma and PeM15 decreased.

#### 3.3.3. Changes in the Intestinal Bacterial Phylotypes

LEfSe was employed to analyze the differential abundance of bacterial taxa in the three groups. In the cladogram, three bacterial taxa contributed to the CK group, including Chitinophagaceae, Campylobacteria, and Oscillospirales; one bacterial taxon contributed to the MC group, namely, Caulobacterales, and eight bacterial taxa contributed to the MCAX group, including Microtrichaceae, Mycobacteriaceae, Ignavibacteria, Polyangia, Rhodocyclaceae, Sutterellaceae, Diplorickettsiales, and Verrucomicrobiales (Figure 6). Among the bacterial genera with an LDA score greater than 3.0, Oscillospirales, Ruminococcaceae, Campylobacterales, Campylobacteria, Campilobacterota, and Chitinophagaceae were dominant in the CK group, Caulobacteraceae and Caulobacterales were dominant in the MC group, and Mycobacteriaceae, Mycobacterium, Sutterellaceae, Verrucomicrobiales, DEV007, and Myxococcota were dominant in the MCAX group (Figure 7).

## 4. Discussion

Cyanobacterial blooms and their microcystins (MCs) have always been a significant problem that threatens the healthy development of the aquaculture industry. MCs are highly carcinogenic, teratogenic, and immunosuppressive, often resulting in a decline in the yield and quality of cultured organisms [10]. Additionally, the consumption of MC-contaminated food poses a potential threat to human health [26]. In the culture of *P*. *vannamei*, the high amount of feeding leads to a high degree of eutrophication in the body of water, exacerbating the outbreaks of cyanobacterial blooms. This has a significant impact on the health of farmed *P*. *vannamei* [27]. Previous research has demonstrated that astaxanthin can improve the growth performance of *P*. *vannamei* to a certain degree [28] and enhance their resistance to pathogenic bacteria [18]. However, the protective effects of astaxanthin on MC-harmed aquatic organisms has only been investigated in crayfish (Procambarus clarkii) [29,30], and no study has examined whether it has a similar effect on *P*. *vannamei*. In the toxicological study of microcystins (MCs) on the *P*. *vannamei*, two main exposure routes were utilized: intraperitoneal injection and immersion [31]. However, there is a lack of research on the effects of low concentrations of MCs in food, particularly regarding their toxic and detoxifying effects. Given that the shrimp species *P*. *vannamei* is omnivorous and feeds on algae, the addition of feed can better simulate the impact of MCs on shrimp in natural aquatic environments. Surprisingly, no studies have been conducted on the potential role of astaxanthin in mitigating the toxic effects of microcystins on *P*. *vannamei*.

Growth performance is a crucial metric for assessing the well-being of aquatic animals, typically quantified as the rate of weight gain. The weight-gain rate of *P*. *vannamei* in the MC group was significantly lower compared to that of the control group, indicating that prolonged consumption of diets containing MCs substantially impeded the growth of *P*. *vannamei*. Other studies have also reported similar inhibitory effects. Dong et al. [32] discovered a significant reduction in the growth rate of fish when diets containing cyanobacteria were fed to juvenile carp and hybrid sturgeon. The growth of bighead carp (Hypophthalmythys nobilis) larvae was significantly inhibited after 10 days of exposure to 30 μg/g MC-LR [33]. Additionally, MC-LR stress led to slow growth and a reduced survival rate in shrimp [34]. Previous studies have demonstrated that adding astaxanthin to the feed can significantly improve the growth performance and low-oxygen tolerance of *P*. *vannamei* [35]. Furthermore, the addition of an appropriate amount of astaxanthin to the feed can significantly promote the growth of both *P*. *vannamei* and *Penaeus japonicus* [36,37]. In this study, it was observed that the average mass and the rate of weight gain of *P*. *vannamei* fed with both MCs and astaxanthin were higher compared to the MC group. These findings indicate that astaxanthin plays a significant role in mitigating the adverse effects of MCs in shrimp cultivation.

The intestinal tract serves as the primary organ for digestion and absorption in *P. vannamei*, and it is also the target organ for MCs. A previous study demonstrated that even trace amounts of MCs (5 mg/kg) in the culture water could lead to significant disruption of the crayfish’s intestinal morphology and abnormal infiltration of myenteric and lamina propria lymphocytes in certain parts of the intestine [38]. Another study observed alterations in the intestinal histology of zebrafish (*Danio rerio*) following exposure to MC-LR [39]. Our current study revealed severe damage to the intestinal histology of *P. vannamei* that were fed MCs. The shrimp’s intestinal epithelial cells were completely detached from the basement membrane, resulting in tissue loss. The disruption of the intestinal tissue structure inevitably affects the digestion and absorption of feed, which is consistent with the aforementioned decline in growth performance. Conversely, the intestinal tissue morphology of shrimp in the astaxanthin-added group exhibited greater integrity. The findings demonstrated that astaxanthin exerted a protective effect on the gut structure. This may be due to the strong antioxidant capacity of AX, which functions to alleviate tissue apoptosis caused by MC-induced oxidative stress [40].

The intestinal barrier is a protective structure that prevents the passage of harmful substances, including bacteria and algal toxins, from the intestinal wall into the body’s tissues, organs, and microcirculation. Pathological conditions, such as trauma, stress, and inflammation, can disrupt the intestinal barrier to different extents, exacerbating the underlying pathology. The intestinal barrier acts as a barrier between colonizing microorganisms and systemic tissues, preventing the invasion of pathogenic bacteria [41]. Gut microorganisms are integral components of the gut barrier in organisms, forming a complex microbial ecosystem consisting of various species of microbiota that rely on each other and are subject to constraints. They actively contribute to the healthy growth of organisms, regulate various life-functions, and prevent the invasion of pathogenic bacteria [42,43,44]. Numerous studies have confirmed that MC is continuously enriched in aquatic organisms through the food chain in the water column and enters the organism most predominantly by ingestion, and is therefore the first to be enriched in the gut and to disrupt the intestinal tissue structure [45,46]. However, no studies have been made on the protective effect of astaxanthin on MC-harmed intestinal microbiota of *P. vannamei*. In this study, we examined the impact of MCs and astaxanthin on the composition, abundance, and diversity of the intestinal microbiota in *P. vannamei*. Previous research has demonstrated that the main intestinal microbiota of shrimp at the phylum level include Ascomycetes, Anaplasma, Thick-walled, Clostridium, and Actinomycetes, with Ascomycetes being the most dominant phylum [38]. Our experiment yielded similar results, with Ascomycetes being the predominant phylum in the intestinal tract of *P. vannamei* in both the control and test groups. This finding aligns with previous studies on aquatic crustaceans. Furthermore, the relative abundance levels of Ascomycetes and Cyanobacteria were significantly higher in the MC group compared to the control group, while the relative abundance levels of Firmicutes and Actinobacteria were lower. These bacterial phyla have consistently been observed in previous studies investigating the effects of MCs on the intestinal microbiota of aquatic animals [47]. The dysbiosis of the intestinal microbiota can be reflected to some extent by the relative abundance of Proteobacteria [48]. In comparison to the MC group, the MCAX group exhibited a significant decrease in the abundance of Proteobacteria in the intestinal microbiota of shrimp, while the relative abundance of Firmicutes increased. Previous studies have demonstrated that astaxanthin can effectively alleviate dextran-sulfate-sodium-induced colitis, preserve the structural integrity of colonic tissues, and promote the growth of Firmicutes in the intestinal tract [49], determinations which align with the findings of the current study.

In this study, we observed significant differences at the genus level between the MC and MCAX groups, particularly in genera associated with host health. We found that the MC group had an increased abundance of conditionally pathogenic bacteria, such as *Vibrio* [50] and *Gemmobacter*, while the MCAX group had a decreased abundance of *Shewanella* [51] and *Vibrio*. After isolation and purification, sequencing, and characterization, it was found that these *Vibrio* spp. were mainly *Vibrio alginolyticus*. This suggests that exposure to MC-LR may have a significant impact on the homeostasis of intestinal microorganisms in *P. vannamei*, leading to an increase in harmful bacteria. Additionally, the addition of astaxanthin altered the abundance of certain bacterial genera in the gut microbiota. We observed that AX attenuates the stress response caused by environmental stresses on shrimp, which contributes to the reduction of intestinal damage and improves growth rates. Therefore, we propose that exposure to MC-LR may disrupt the composition and stability of the gut microbiota, potentially impairing the gut health of the host. However, astaxanthin had a significant modulatory effect on the imbalance of the gut microbiota in MC-affected *P. vannamei*, and it was able to reduce the levels of some unhealthy microbiota.

## 5. Conclusions

This study aimed to examine the protective effects of astaxanthin and its mechanism given the addition of MC-LR to *P. vannamei*. The results suggested that astaxanthin has the potential to enhance the growth of *P. vannamei* by preserving the integrity of the intestinal tract and modifying the composition of the intestinal microbiota. The present study demonstrated that astaxanthin can mitigate the detrimental effects of MC on *P. vannamei* aquaculture. This study proposes technical measures to minimize the loss of aquaculture caused by MC.

## Figures and Tables

**Figure 1 animals-14-00058-f001:**
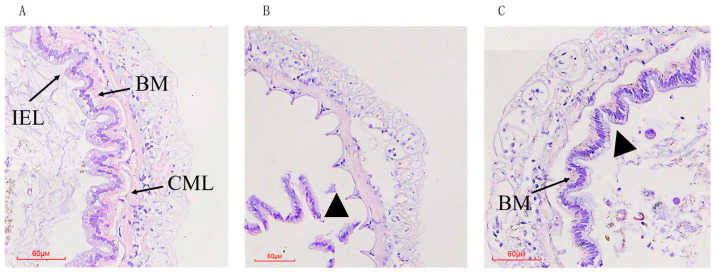
Histological analysis of intestine of *P. vannamei* fed from 15 days with either the control diet (**A**), a diet containing MC-LR (**B**), or a diet containing MC-LR and AX (**C**). BM: basement membrane; CML: circular muscle layers; IEL: intestinal epithelial layer. Triangle: multifocal necrosis with loss of tissue. Stained with hematoxylin and eosin (H/E), 200×.

**Figure 2 animals-14-00058-f002:**
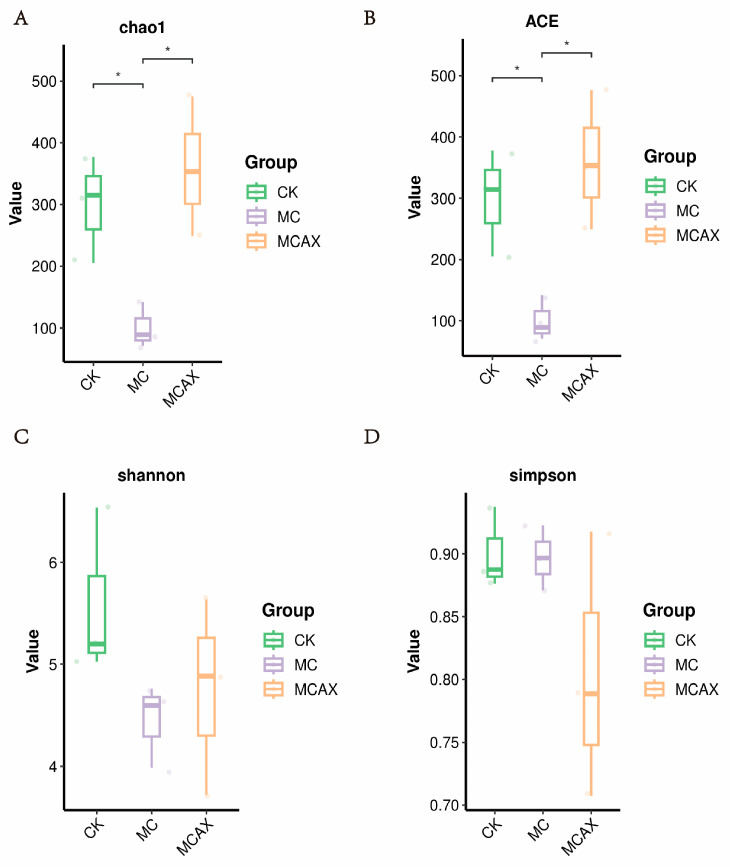
Comparison of gut microbial diversities and composition of the three groups. Gut bacterial alpha diversity can be determined by Chao1 (**A**), ACE (**B**), Shannon (**C**) and Simpson (**D**). The data are expressed as the Mean ± SD. * *p* < 0.05.

**Figure 3 animals-14-00058-f003:**
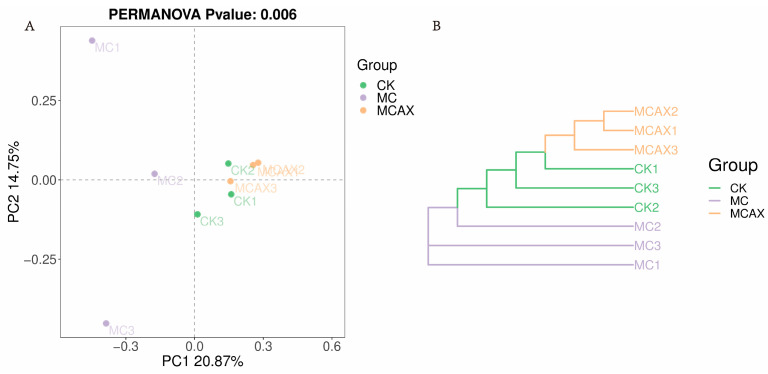
Scatterplot from PCoA on the basis of binary Jaccard distance in the gut bacterial community (**A**). The clustering analysis is based on a binary Jaccard pair-group method with arithmetic means (UPGMA) (**B**).

**Figure 4 animals-14-00058-f004:**
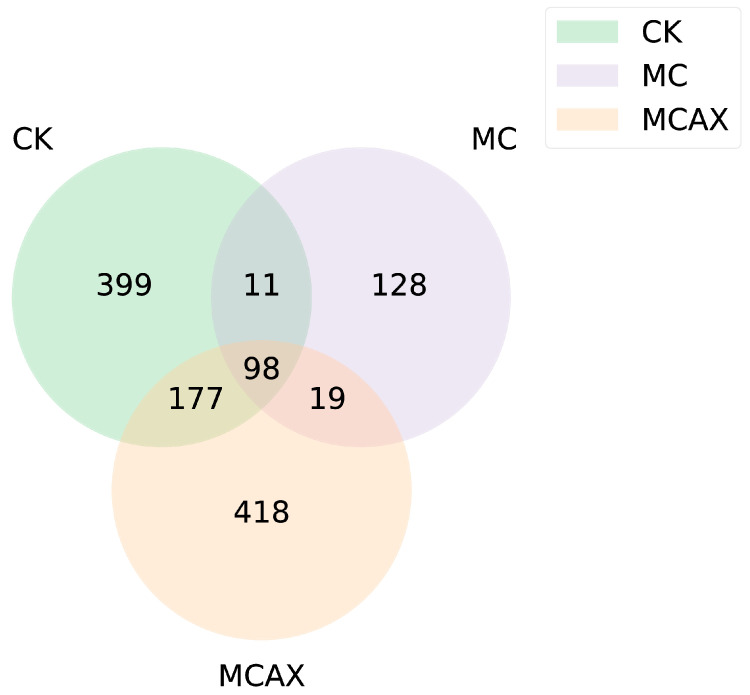
Venn diagram showing the unique and shared ASV among the three groups. CK represents the control group, MC represents the MC group, and MCAX represents the MC + AX group.

**Figure 5 animals-14-00058-f005:**
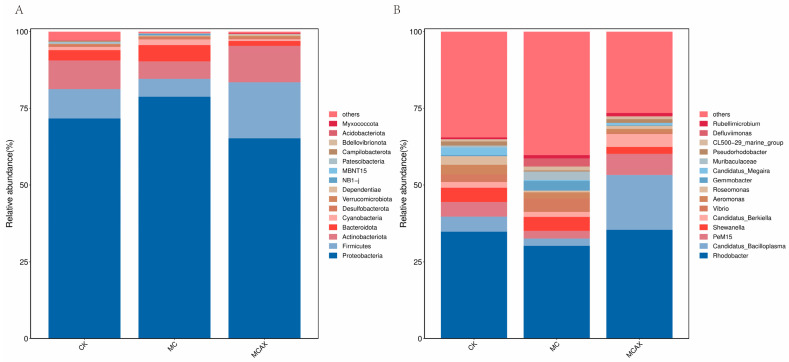
Relative abundance of the intestinal microbiota of *P. vannamei*. (**A**) The bar plots at phylum level. (**B**) The bar plots at genus level. CK represents the control group, MC represents the MC group, and MCAX represents the MCAX group.

**Figure 6 animals-14-00058-f006:**
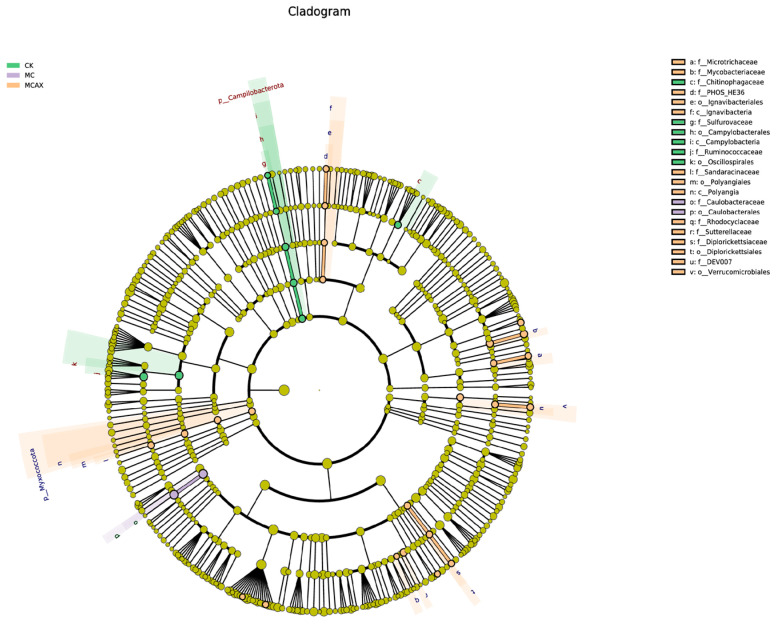
The LEfSe analysis of the differential microbiota of three groups. Differently colored nodes are used to indicate microbial taxa that exhibit significant enrichment in the corresponding group and have a notable impact on intergroup differences. Yellowish nodes represent microbial taxa that do not show significant differences among any of the different subgroups or do not have a significant effect on intergroup differences.

**Figure 7 animals-14-00058-f007:**
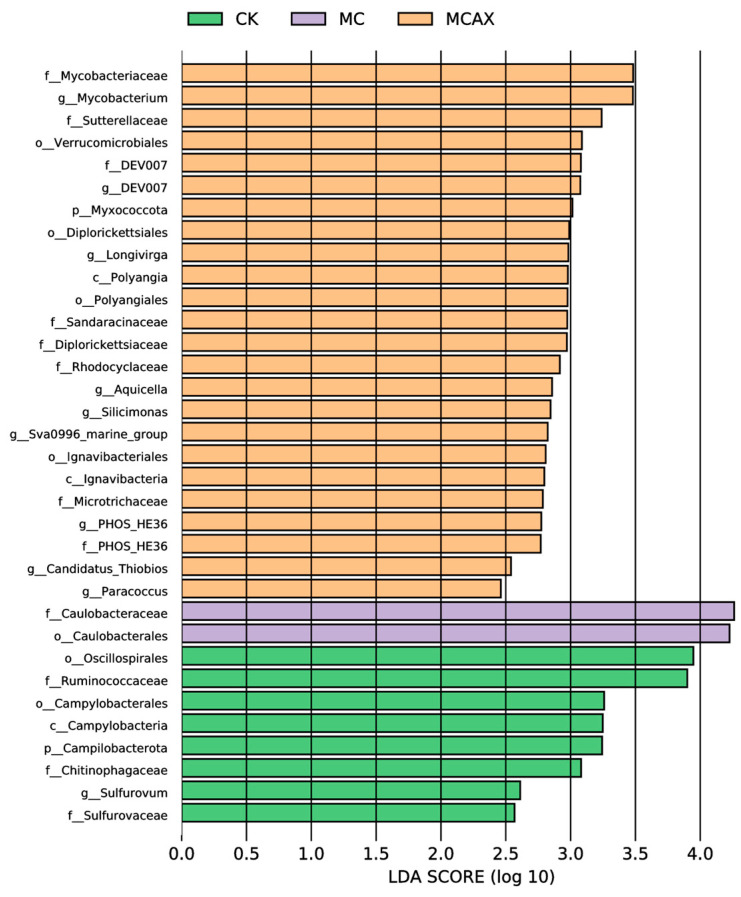
The LDA scores of different microbes of three groups. The LDA scores were obtained through LDA analysis, in which larger scores indicate a stronger influence of species abundance on differential effects.

**Table 1 animals-14-00058-t001:** Compositions of different *P. vannamei* feeds.

Items	Transient Periods (7 d)	Experimental Periods	
		Pre-Toxin Attack (30 d)	Toxin Attack (15 d)
**CK**	Base feed	Base feed	Base feed
**MC**	Base feed	Base feed	Base feed + MC-LR (100 μg/kg)
**MCAX**	Base feed	Base feed + AX (100 mg/kg)	Base feed + AX (100 mg/kg) + MC-LR (100 μg/kg)

**Table 2 animals-14-00058-t002:** Growth performance of *P. vannamei* fed with different experimental diets for 15 days (mean of triplicates ± SD).

Items	CK	MC	MCAX
Initial weight (g)	7.80 ± 0.01	7.78 ± 0.01	7.79 ± 0.01
Final weight (g)	9.36 ± 0.23 ^a^	8.96 ± 1.21 ^b^	9.12 ± 0.33 ^a^
PWG (%)	16.64 ± 0.34 ^a^	13.44 ± 0.25 ^b^	14.61 ± 0.22 ^c^
SGR (%)	2.24 ± 0.12 ^a^	0.77 ± 0.10 ^b^	2.21 ± 0.16 ^a^
Survival rate (%)	96.78 ± 4.45	92.50 ± 3.56	95.34 ± 2.78

Values in the same row with different superscripts are significantly different (*p* < 0.05).

## Data Availability

Follow-up research on this project is ongoing; please contact the corresponding author with reasonable requests.

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
