# Peer review of "Effects of Astaxanthin on Growth Performance, Gut Structure, and Intestinal Microorganisms of Penaeus vannamei under Microcystin-LR Stress"

_animals, 2023, doi:10.3390/ani14010058_

Round 1
Reviewer 1 Report
Comments and Suggestions for Authors
Journal: Animals (ISSN 2076-2615)
Manuscript ID: animals-2753257
Type: Article
Title: Effects of Astaxanthin on Growth Performance, Gut Structure and Intestinal Microorganisms of Penaeus vannamei under Microcystin-LR Stress
Section: Aquatic Animals
This study aimed to investigate the protective effect of astaxanthin (AX) on Penaeus vannamei, which was stimulated by Microcystin-LR (MC-LR) through analyzing their growth performance, intestinal structure, and intestinal flora.
Before this work will be recommended or will be given any possible acceptance few comments must be incorporated for improving the quality of this work as well as for further publication in this reputed journal. I have the following observations or queries and comments which may further enhance your piece of work. The authors require to modify the following points in detail.
1. Simple Summary
Lines 10-12
“This study aimed to investigate the initial protective effect of astaxanthin (AX) on Penaeus vannamei, which was stimulated by Microcystin-LR (MC-LR) through analyzing their growth performance, intestinal structure, and intestinal flora.”
“initial protective effect of astaxanthin (AX)”, what do you mean by “initial”?
2. Abstract
Lines 27-32
“After 15 days of feeding, the specific growth rate (SGR) was significantly higher in the MCAX group (2.21% day-1) compared to the MC group (0.77% day-1), and there was no significant difference between the MCAX group (2.21% day-1) and the control group (2.24% day-1). Similarly, the overall weight gain (OWG) was also significantly higher in the MCAX group (14.61% ) compared to the MC group (13.44%) and the control group (16.64%). Furthermore, the intestinal morphology was found to be more intact in the MCAX group compared to the MC group.”
What about results of MCAX and MC groups, when compared with the control group? Please revise the abstract.
3. Section “1. Introduction”
Lines 62-64
MCs can impact the growth rate of aquatic animals, leading to a decrease in shrimp growth and causing pathological changes in shrimp organs, including the hepatopancreas, intestines, and gonads.”
Please read and cite the following paper.
Challenges of using blooms of Microcystis spp. in animal feeds: A comprehensive review of nutritional, toxicological and microbial health evaluation. https://doi.org/10.1016/j.scitotenv.2020.142319.
4. Lines 69-71
“Studies have confirmed that astaxanthin promotes the growth, survival, and coloration of cultured subjects.”
Please insert the references.
5. Section “2. Materials and Methods”
Line 84
“MC-LR is supplied by Shanghai Yuanye Biotechnology Co., Ltd.”
Does Shanghai Yuanye Biotechnology Co., Ltd. produce MC-LR or this company sell MC-LR produced by other company? Please provide more details.
6. Lines 88-91
“They are dissolved in a ratio of 1:3 with water to prepare the feed with an MC-LR content of 100 μg/kg and an AX content of 100 mg/kg. The control group only receives water without the addition of AX and MC-LR. The feed is naturally dried under normal conditions and stored at 4℃.”
What are “They”? 100 μg of MC-LR and AX per wet weight or dry weight kg of feed? Please make it clear in the revised manuscript. Please also check the entire manuscript.
7. Lines 107-113
“The control and treatment groups (MC) were provided with a basal diet, while the treatment group (MCAX) received a diet containing 100 mg/Kg of AX. The MCAX group was fed this diet for a duration of 30 days. At the conclusion of the 30th day, the control group maintained their consumption of the basal diet. The MC group received a diet enriched with MC-LR (100 μg/Kg), while the MCAX group received a diet containing both MC-LR (100 μg/Kg) and AX (100 mg/Kg) for a duration of 15 days.”
How long is the treatment duration? 30 days or 15?
7. Section “2.6. DNA extraction and amplification”
How did you treat the intestine for DNA extraction? How many shrimps did you use? Please insert more details in the revised manuscript.
8. Section “2.7. Library construction and sequencing”
What was the sequencing size for each sample? Please insert more details in the revised manuscript.
9. Section “3. Results”
overall weight gain (OWG), Percent weight gain (PWG). Please unify the item.
10. Figure 1
Please insert the scale bar in each histological image to show the size of tissues.
11. Figure 6A, B
The words in the figure are too small. Please increase the font of words.
12. Section “4. Discussion”
Lines 269-272
“Cyanobacterial blooms and their microcystins (MCs) have always been a significant problem that threatens the healthy development of the aquaculture industry. MCs are highly carcinogenic, teratogenic, and immunosuppressive, often resulting in a decline in the yield and quality of cultured organisms.”
Please read and cite the following paper.
Challenges of using blooms of Microcystis spp. in animal feeds: A comprehensive review of nutritional, toxicological and microbial health evaluation. https://doi.org/10.1016/j.scitotenv.2020.142319.
13. How did Microcystin-LR (MC-LR) and astaxanthin (AX) affect the intestinal structure, and intestinal flora of Penaeus vannamei? What are the mechanisms? Did Microcystin-LR (MC-LR) and astaxanthin (AX) enter the gut?
Comments on the Quality of English Languagemoderate edits
Author Response
Dear reviewers:
We feel great thanks for your professional review work on our article. As you have pointed out, there are several issues that need to be addressed. Based on your valuable suggestions, we have made extensive revisions to our previous draft. The detailed corrections are listed below.
- Simple Summary
Lines 10-12
“This study aimed to investigate the initial protective effect of astaxanthin (AX) on Penaeus vannamei, which was stimulated by Microcystin-LR (MC-LR) through analyzing their growth performance, intestinal structure, and intestinal flora.”
“initial protective effect of astaxanthin (AX)”, what do you mean by “initial”?
Response : Thank you for your valuable comments . When we refer to the “initial protective effect” of astaxanthin (AX), it means that we are examining the immediate or early-stage impact of AX on the organism under investigation. If you think this expression is not accurate enough, I can delete the word ‘’initial’’, thank you!
- Abstract
Lines 27-32
“After 15 days of feeding, the specific growth rate (SGR) was significantly higher in the MCAX group (2.21% day-1) compared to the MC group (0.77% day-1), and there was no significant difference between the MCAX group (2.21% day-1) and the control group (2.24% day-1). Similarly, the overall weight gain (OWG) was also significantly higher in the MCAX group (14.61% ) compared to the MC group (13.44%) and the control group (16.64%). Furthermore, the intestinal morphology was found to be more intact in the MCAX group compared to the MC group.”
What about results of MCAX and MC groups, when compared with the control group? Please revise the abstract.
Response: Thanks for your valuable advice. Supplementary information has been included in the manuscript abstract when comparing the results of the MCAX and MC groups to the control group. (L32-35)
- Section “1. Introduction”
Lines 62-64
MCs can impact the growth rate of aquatic animals, leading to a decrease in shrimp growth and causing pathological changes in shrimp organs, including the hepatopancreas, intestines, and gonads.”
Please read and cite the following paper.
Challenges of using blooms of Microcystis spp. in animal feeds: A comprehensive review of nutritional, toxicological and microbial health evaluation. https://doi.org/10.1016/j.scitotenv.2020.142319.
Response: Thanks for your valuable advice. We have read and cited this paper.( References:10)
- Lines 69-71
“Studies have confirmed that astaxanthin promotes the growth, survival, and coloration of cultured subjects.”
Please insert the references.
Response: Thanks for your valuable advice. We have inserted the references.( References:16)
- Section “2. Materials and Methods”
Line 84
“MC-LR is supplied by Shanghai Yuanye Biotechnology Co., Ltd.”
Does Shanghai Yuanye Biotechnology Co., Ltd. produce MC-LR or this company sell MC-LR produced by other company? Please provide more details.
Response: Thank you for pointing this out and again I apologize for my carelessness. MC-LR was produced by Shanghai Yuanye Biotechnology Co., Ltd. The purity of MC-LR is 95%. The manuscript has been revised.(L91 )
- Lines 88-91
“They are dissolved in a ratio of 1:3 with water to prepare the feed with an MC-LR content of 100 μg/kg and an AX content of 100 mg/kg. The control group only receives water without the addition of AX and MC-LR. The feed is naturally dried under normal conditions and stored at 4℃.”
What are “They”? 100 μg of MC-LR and AX per wet weight or dry weight kg of feed? Please make it clear in the revised manuscript. Please also check the entire manuscript.
Response: Thanks for the heads up, "They" refers to MC-LR and AX, which we have made it clear in the revised manuscript. (L98-100)
- Lines 107-113
“The control and treatment groups (MC) were provided with a basal diet, while the treatment group (MCAX) received a diet containing 100 mg/Kg of AX. The MCAX group was fed this diet for a duration of 30 days. At the conclusion of the 30th day, the control group maintained their consumption of the basal diet. The MC group received a diet enriched with MC-LR (100 μg/Kg), while the MCAX group received a diet containing both MC-LR (100 μg/Kg) and AX (100 mg/Kg) for a duration of 15 days.”
How long is the treatment duration? 30 days or 15?
Response: Thank you for pointing this out, In order to describe the experimental process more effectively, we divided the experimental period into three stages in chronological order: transient, pre-toxin attack, and toxin attack stages (Table 1). Therefore, the treatment duration should be 45 days. A detailed description has been provided in the manuscript.(L1.3-104)
- Section “2.6. DNA extraction and amplification”
How did you treat the intestine for DNA extraction? How many shrimps did you use? Please insert more details in the revised manuscript.
Response: Thank you for notifying us. The intestines of six shrimp were selected from each tank were mixed for one mi-crobial sample, and each group included three replicate microbial samples. we have included this information in the revised manuscript.(L148-150 )
- Section “2.7. Library construction and sequencing”
What was the sequencing size for each sample? Please insert more details in the revised manuscript.
Response: Thank you for notifying us. The sequencing size for each sample is 250bp, and we have included this information in the revised manuscript.(L166 )
- Section “3. Results”
overall weight gain (OWG), Percent weight gain (PWG). Please unify the item
Response: Thank you for pointing this out and again I apologize for my carelessness, and we have unified this item in the revised manuscript.( L200)
- Figure 1
Please insert the scale bar in each histological image to show the size of tissues.
Response: Thank you for pointing this out, and we have inserted a scale bar in each histological image in the revised manuscript. ( L211 )
- Figure 6A, B
The words in the figure are too small. Please increase the font of words.
Response: Thank you for pointing this out, and we have increased the font of words in the figure 6A,B in the revised manuscript. ( L282-283 )
- Section “4. Discussion”
Lines 269-272
“Cyanobacterial blooms and their microcystins (MCs) have always been a significant problem that threatens the healthy development of the aquaculture industry. MCs are highly carcinogenic, teratogenic, and immunosuppressive, often resulting in a decline in the yield and quality of cultured organisms.”
Please read and cite the following paper.
Challenges of using blooms of Microcystis spp. in animal feeds: A comprehensive review of nutritional, toxicological and microbial health evaluation. https://doi.org/10.1016/j.scitotenv.2020.142319.
Response: Thanks for your valuable advice. We have read and cited this paper. ( References:10 )
13. How did Microcystin-LR (MC-LR) and astaxanthin (AX) affect the intestinal structure, and intestinal flora of Penaeus vannamei? What are the mechanisms? Did Microcystin-LR (MC-LR) and astaxanthin (AX) enter the gut?
Thank you for your question. This study found that Microcystin-LR (MC-LR) and astaxanthin (AX) have an impact on the intestinal structure and intestinal microbiota of Penaeus vannamei, with possible mechanisms:
Regarding the intestinal structure, MC-LR can cause cell death and inflammatory reactions, resulting in damage to the intestinal mucosa. On the other hand, AX, as a natural antioxidant, has a protective effect on cell membranes against oxidative damage, which may help alleviate the adverse effects of MC-LR on the intestinal structure.
Regarding the intestinal microbiota, MC-LR may disrupt the balance of Penaeus vannamei’s intestinal microbiota, leading to decreased microbial diversity and reduced numbers of certain beneficial bacteria. On the other hand, AX can enhance the immune system and inhibit the growth of harmful bacteria, thus protecting the balance of the intestinal microbiota.
Some studies have shown that MC-LR can enter the digestive tract through oral intake and further be absorbed and accumulated in the intestines. As a natural compound, AX is typically added to feed and can enhance the coloration of muscles, indicating that it can be absorbed in the intestines.
Reviewer 2 Report
Comments and Suggestions for Authors
Overall opinion
The manuscript "Effects of Astaxanthin on Growth Performance, Gut Structure and Intestinal Microorganisms of Penaeus vannamei under Microcystin-LR Stress" by Guolin Song et al. describes experimental results on the supplementation of Penaeus vannamei diets with a carotenoid, astaxanthin (AX). P. vannamei is a commercially valuable species cultivated worldwide and is susceptible to microcystin toxicity. AX is thought to be able to mitigate microcystin toxicity. It was found that MC and MCAX groups exhibited differences in intestinal morphology (more intact in MCAX group) and microflora composition (decrease in both Shewanella and Vibrio abundance in MCAX group), which was attributed to the MC-LR mitigating effect of AX via preserving the structural integrity of the intestinal mucosa and regulating the intestinal flora composition.
The manuscript is in line with the journal's topics. The content could be of interest to both basic researchers and practitioners (fish farmers, feed producers). However, I have some crucial points to reject this MS due to methodological concerns.
Major concerns
It is well known that astaxanthin is practically insoluble in water and other polar solvents. You write that astaxanthin and microcystin "are dissolved in a ratio of 1:3 with water to prepare the feed". Was it taken into account when calculating the dosage of AX (which is not indicated in the text, but only its final concentration in the feed - 100 mg AX per kg of feed) that commercial AX "contains at least 10% AX"? It turns out that 1 g of the commercial preparation of AX should have been dissolved in such a small amount of solvent as indicated - 3 g (ml) of water?
Minor concerns
Abstract.
Lines 17-39, Please, revise the whole text of abstract to exclude inappropriate line breaks.
Materials and Methods
Line 90, The phrase “The control group only receives water without the addition of AX and MC-LR” is indeed confusing. Water was added to the feed, which was then dried, wasn't it?
Line 107, Better write “The control and treatment (MC) groups” than “The control and treatment groups (MC).
Line 123, Please specify the methodological precautions taken during dissecting the intestines.
Figures.
Figure 6. The text in a legend on the right (B) is illegible. The caption is separate from the figure.
Author Response
Dear reviewers:
We feel great thanks for your professional review work on our article. As you have pointed out, there are several issues that need to be addressed. Based on your valuable suggestions, we have made extensive revisions to our previous draft. The detailed corrections are listed below.
Comments 1:
Major concerns
It is well known that astaxanthin is practically insoluble in water and other polar solvents. You write that astaxanthin and microcystin "are dissolved in a ratio of 1:3 with water to prepare the feed". Was it taken into account when calculating the dosage of AX (which is not indicated in the text, but only its final concentration in the feed - 100 mg AX per kg of feed) that commercial AX "contains at least 10% AX"? It turns out that 1 g of the commercial preparation of AX should have been dissolved in such a small amount of solvent as indicated - 3 g (ml) of water?
Response: Thank you for pointing this out, Astaxanthin has three sources: algal, yeast, and chemically synthesized. The first two sources are lipids, which are insoluble in water, as you mentioned. In this study, we used the chemically synthesized astaxanthin, which was synthesized with hydrophilic groups and encapsulated with hydrophilic materials. This form of astaxanthin can dissolve well in cold water, is relatively inexpensive, and is commonly used to enhance the color of salmon and ornamental fish. We had multiple communications with the manufacturer prior to its use.
Comments 2:
Abstract.
Lines 17-39, Please, revise the whole text of abstract to exclude inappropriate line breaks.
Response: Thank you for pointing this out, we have excluded inappropriate line breaks throughout the entire abstract in the manuscript.(L17-45 )
Comments 3:
Materials and Methods
Line 90, The phrase “The control group only receives water without the addition of AX and MC-LR” is indeed confusing. Water was added to the feed, which was then dried, wasn't it?
Response:
Yes, no AX and MC-LR were added to the control group, and only water was added in order to maintain the same water content in all three groups of feeds, which made it easier to weigh the feeds at a later stage of feeding, and to minimize experimental errors. Thanks again.
Comments 4:
Line 107, Better write “The control and treatment (MC) groups” than “The control and treatment groups (MC).
Response: Thanks for your valuable advice. We have used “The control and treatment (MC) groups” instead of “The control and treatment groups (MC)’’. Other locations in the manuscript have been modified (L121 )
Comments 5:
Line 123, Please specify the methodological precautions taken during dissecting the intestines.
Response: Thank you for pointing this out. We have specified the methodological precautions taken during the dissection of the intestines and included it in the revised manuscript under DNA extraction and amplification.(L148-150)
Comments 6:
Figures.
Figure 6. The text in a legend on the right (B) is illegible. The caption is separate from the figure.
Response: Thank you for pointing this out, and we have increased the font of words in the figure 6A,B in the revised manuscript.
Reviewer 3 Report
Comments and Suggestions for Authors
Most of my comments are included in the edited PDF of the submitted version.
The main weakness of the study is that the authors do not consider at all the protective mode of action of astaxanthin in front of MC, as well as the toxicity mechanisms of MC affecting growth performance, gut histological organization, and microbiota. This is needed to increase the overall soundness of the manuscript and its potential impact.

Comments on the Quality of English Languagethe use of some English words needs to be checked
Author Response
Dear reviewers:
We feel great thanks for your professional review work on our article. As you have pointed out, there are several issues that need to be addressed. Based on your valuable suggestions, we have made extensive revisions to our previous draft.

Reviewer 4 Report
Comments and Suggestions for Authors
Effects of Astaxanthin on Growth Performance, Gut Structure 2 and Intestinal Microorganisms of Penaeus vannamei under Microcystin-LR Stress was studied in this paper
The abstract is too long, I suggest reducing it by a few lines.
Space compresses from pages 265, 266, 267
Author Response
Dear reviewers:
We feel great thanks for your professional review work on our article. As you have pointed out, there are several issues that need to be addressed. Based on your valuable suggestions, we have made extensive revisions to our previous draft. The detailed corrections are listed below.
Comments 1: The abstract is too long, I suggest reducing it by a few lines.
Response: Thanks for your valuable advice. We've simplified the abstract.
Comments 2: Space compresses from pages 265, 266, 267
Response: Thanks for your valuable advice. We’ve compressed space and revised it in the revised manuscript.
Round 2
Reviewer 1 Report
Comments and Suggestions for Authors
Journal: Animals (ISSN 2076-2615)
Manuscript ID: animals-2753257-peer-review-v2
Type: Article
Title: Effects of Astaxanthin on Growth Performance, Gut Structure and Intestinal Microorganisms of Penaeus vannamei under Microcystin-LR Stress
Section: Aquatic Animals
This study aimed to investigate the protective effect of astaxanthin (AX) on Penaeus vannamei, which was stimulated by microcystin-LR (MC-LR) through analyzing their growth performance, intestinal structure, and intestinal flora.
The manuscript improved during the revisions. However, before this work will be recommended or will be given any possible acceptance few comments must be incorporated for improving the quality of this work as well as for further publication in this reputed journal. I have the following observations or queries and comments which may further enhance your piece of work. The authors require to modify the following points in detail.
1. Simple Summary
Lines 10-11
“This study aimed to investigate the initial protective effect of astaxanthin (AX) on Penaeus vannamei”
Please delete the “initial”, as AX was provided both before and during MC exposure.
2. Abstract
Lines 33-35
“However, in the MCAX group, although there was still a gap between the intestinal epithelial cells and the basement membrane, the overall intestinal morphology was more intact than in the MC group.”
“the overall intestinal morphology was more intact than in the MC group”, the intestinal morphology was intact in the MCAX group? Was there any damage? Please revise the sentence.
3. Section “2. Materials and Methods”
Lines 95-96
“In every kilogram of feed, 100 mg/kg of AX and 100 μg/kg of MC-LR are added.”
How did you select the doses in this study? Please add some explanations in the revised manuscript.
4. Lines 99-100
“Then, according to the experimental design (Table 1), spray the MC-LR and AX solutions onto the basal feed.”
Please change “spray the MC-LR and AX solutions onto the basal feed” to “the MC-LR and AX solution was sprayed onto the basal feed”.
5. Section “2.7. Library construction and sequencing”
What is the sequence size for each sample? 5G, 10G or …? 30,000, 50,000 sequences or how many for each sample? Please insert the details in the revised manuscript. Also, please submit the sequencing data to NCBI.
6. Section “3. Results”
Lines 206-209
“In the MCAX group, there was a gap between the intestinal epithelial cells and the basement membrane, and the epithelial cell layer was separated from the basement membrane to some extent compared to control group, while the intestinal morphology was more complete than that of the MC group (Fig. 1C).”
“while the intestinal morphology was more complete than that of the MC group”, the intestinal morphology was complete in the MCAX group? Was there any damage? Please revise the sentence.
7. Fig. 5AB
Please insert the error bars.
8. Fig. 6, 7
Please insert figure captions to introduce how to read the figures and the results of figure in details.
9. Section “4. Discussion”
Lines 349-350
“It may be due to the strong antioxidant capacity of AX to alleviate tissue apoptosis caused by MCs-induced oxidative stress.”
Please insert the references.
10. “Vibrio are not always pathogenic, there are some Vibrios with probiotic effects. Please, modulate this sentence, since authors only identified bacteria at genus and not species level.
Response: Thank you for pointing this out, After isolation and purification, sequencing
and characterization, it was found that these Vibrio spp. were mainly Vibrio alginolyticus, and therefore typical conditional pathogens.”
Please insert the details of Vibrio alginolyticus in the revised manuscript.
11. Editing of English language is required.
Comments on the Quality of English Languagemoderate edits
Reviewer 2 Report
Comments and Suggestions for Authors
I acknowledge the extensive work of the authors in improving the text of the MS. However, there are still some rough spots in the text that need to be addressed. Some of these are listed below and the others should be checked throughout the text.
Line 74, In the sentence “Astaxanthin is a carotenoid that is not derived from vitamin A and is known for its strong antioxidant properties [13][14], which are widely found in nature[15]” it is not clear what is “found”: carotenoids? Vitamin A-derived substances? properties? In any case, consistency is required.
Line 93, In “The feed (crude protein=42%, crude fat=4%, crude 93 fiber=5%, ash=16%)” the repeated signs “=” in brackets seem unnecessary.
Line 100, "The control group only receives water" I recommend stating "the control diet was sprayed with water" rather than “the group ... receives".
Lines 141, 148, “The intestines of three shrimps were selected from each tank and transplanted into fixation solution” should be reworded to clarify that “shrimps were selected” whereas “intestines were dissected and fixed” such as “The intestines of three shrimps selected from each tank were dissected and fixed in the solution ...”.
Reviewer 3 Report
Comments and Suggestions for Authors
comments have been addressed by authors.
English language needs to be revised
Comments on the Quality of English LanguageEnglish language needs to be revised